# Model-Based Fault Diagnosis of Actuators in Electronically Controlled Air Suspension System

**Xinwei Jiang, Xing Xu * and Haiqiang Shan**

Automotive Engineering Research Institute, Jiangsu University, 301 Xuefu Road, Zhenjiang 212013, China
* Correspondence: xuxing@ujs.edu.cn

**Abstract:** The air suspension adjusts the height of the vehicle body through charging and bleeding air to meet the high performance of the vehicle, which needs a reliable electronic control system. Through fault tree analysis of the electronically controlled air suspension (ECAS) system and considering the correlation between the duty cycle and flow rate of the air spring solenoid valve, the fault model of the solenoid valve is constructed, and the fault diagnosis design method of the ECAS system solenoid valve based on multiple extended Kalman filter banks (EKFs) is proposed. An adaptive threshold is used to realize fault diagnosis, and active fault-tolerant control is carried out based on an analytical model. The real controller based on d2p rapid prototyping technology and the vehicle model based on AMESim are further verified on the hardware-in-the-loop (HiL) simulation test platform and compared with the pure simulation results. The test results show that the fault diagnosis and fault-tolerant control algorithm can work normally in the actual controller, and can effectively realize the fault diagnosis and fault-tolerant control of the actuator in the vehicle ECAS system.

**Keywords:** electronically controlled air suspension; solenoid valve; extended Kalman filter bank; fault diagnosis; fault-tolerant control

## 1. Introduction

Air suspension can improve vehicle ride comfort and road friendliness, and its natural frequency is low and has variable stiffness characteristics [1–5]. However, the general air suspension cannot adjust the suspension stiffness and damping according to the load change. The natural frequency and controllability of the electronically controlled air suspension (ECAS) are low, which can further improve the vehicle ride comfort and control stability [6–8]. In recent years, research on ECAS has mainly focused on improving comfort and stability. In 2019, Rui modeled the ECAS system according to its nonlinear characteristics and designed an adaptive sliding mode control strategy. The method effectively improves the stability of the system by considering the parameter uncertainty [9]. In 2021, Ma et al. designed an integrated control strategy to solve the problems of small stiffness adjustment range and poor roll stability of traditional ECAS systems. The handling stability and anti-roll performance of the vehicle are improved [10]. In 2021, Hu et al. conducted research on the hybrid control of body height and attitude of the ECAS system. They built a vehicle model based on mixed logic dynamics and designed the switching strategy of the solenoid valve. The coordinated control between the ECAS system body height and attitude is well solved, and good vibration isolation performance and stability are achieved [11].

However, the ECAS system is highly dependent on the reliability of each component. Sensors and actuators are very important components [12,13]. The structure of the ECAS system is shown in Figure 1. The actuator is the air spring solenoid valve. If any of the four solenoid valves fail, the ride comfort and handling stability of the whole vehicle will be severely affected [14]. Therefore, it is necessary to consider the reliability of the actuator of the electronically controlled air suspension system.

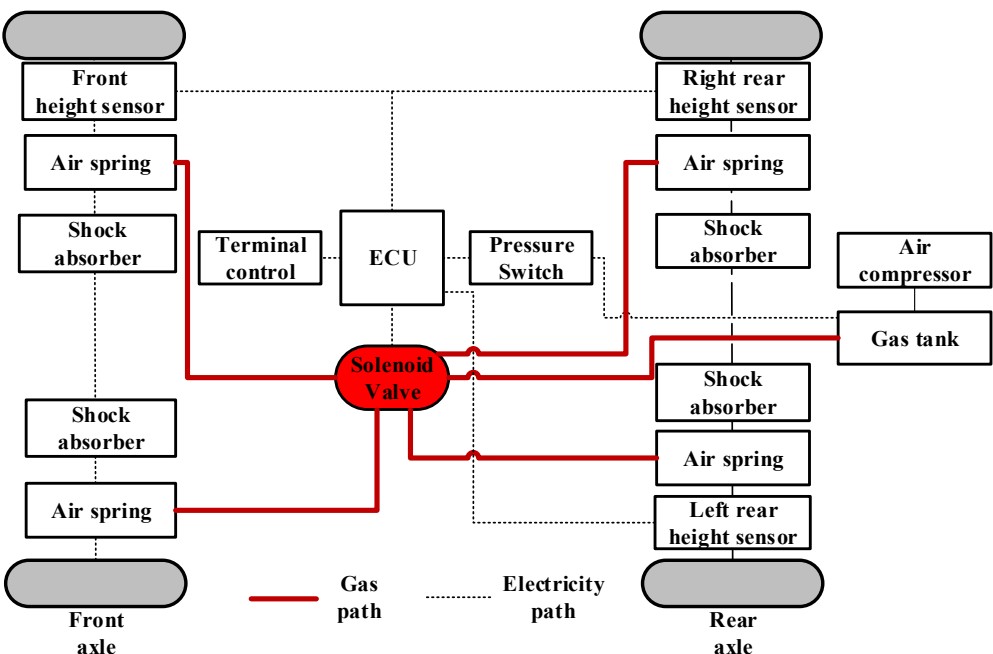

**Figure 1.** ECAS system.

The fault diagnosis technology in the reliability research of the control system is widely used in the control system of various industries. The technology can effectively improve the reliability and maintainability of the system. The technology was first proposed in the 1990s by Frank, who divided the fault diagnosis methods into three categories based on an analytical model, expert knowledge, and signal processing [15]. In 2009, Zhou further developed the method from qualitative and quantitative perspectives [16]. In 2020, Feng proposed an SVM model and effectively distinguished different fault conditions of trains by using the support vector machine method [17]. The fault diagnosis method based on qualitative analysis is mainly divided into the graph theory method, the expert system method, and the qualitative simulation method. Among them, the graph theory method includes the symbol-directed graph method [18] and the fault tree method [19]. The main principle is to judge the fault according to the logical causal relationship. This method can be understood easily and is widely adopted.

The fault location and type can be determined by fault diagnosis. In this way, fault-tolerant control (FTC) can be implemented. Alwi et al. classified fault-tolerant control methods in detail [20]. Fault-tolerant control is usually divided into passive FTC (PFTC) and active FTC (AFTC). The common methods of passive fault-tolerant control include H methods based on ∞ control theory [21] and sliding mode control theory [22]. The characteristic of passive fault-tolerant control is that there is no need for fault diagnosis, and the controller parameters are not changed, so it is easy to implement, but the fault-tolerant control is limited. The active fault-tolerant control adjusts the controller parameters online or configures the controller structure units online based on the fault diagnosis information to realize the system stability, which is different from the passive fault-tolerant control. Active fault-tolerant control methods can be divided into two types: planning type and online adjustment type [23]. In the planning of FTC, the controller is designed in advance for all possible faults of the system, and the corresponding controller is activated when the corresponding fault occurs. On the other hand, the online adjustment of the controller is mainly through adaptive control or control signal redistribution to achieve fault-tolerant control [24,25]. In 2020, Pang et al. designed a fault-tolerant controller based on the nonlinear suspension system, which combines the state feedback observer and the H-infinity observer. The purpose of fault-tolerant control is achieved by comparing the system state under the fault-free state to compensate [26]. In 2021, Xue et al. considered the failure of a 1/4 active suspension actuator under the change of sprung mass. They used



the finite-element neural network method to approximate the observer fault, and turned it into a linear matrix inequality problem. This method can obtain the results faster and more accurately, and can well adapt to the changing spring mass [27]. The above methods are very effective, but the actuator fault diagnosis and fault-tolerant control for the ECAS system are rarely used. In order to fill the shortage in this field, this research will use the extended Kalman filter bank with an adaptive threshold as the observer, and use the online adjustment method to continuously improve the body height. Compared with the existing research, this method has stronger adaptability and accuracy.

At present, research on the fault diagnosis and fault-tolerant control of the ECAS system is scarce. In addition, research on ECAS vehicle height control rarely focuses on how to ensure effective control in the case of actuator failure. However, the ECAS system with actuator failure has many problems to be solved, such as risk analysis, fault diagnosis architecture, and fault-tolerant control strategy design.

The contribution of this paper is to use the extended Kalman filter bank based on an adaptive threshold to study the fault diagnosis and fault-tolerant control of the air spring solenoid valve. This method can change the threshold according to the system input, and can effectively reduce the probability of missed diagnosis and misdiagnosis. After fault diagnosis and isolation, corresponding active fault-tolerant methods are adopted for different types of faults. Among them, the method of online adjustment of controller parameters is used to improve the height and attitude control effect of the constant gain fault. In addition, a hardware-in-the-loop simulation platform is built to finally test the effectiveness and accuracy of the above methods. The HiL platform can realize the connection between the real controller and the simulation model of the controlled object, so as to form a complete loop to test the actual operation and feasibility of fault diagnosis and fault-tolerant control algorithm model in the real controller. This paper provides a research idea for the operation stability and fault-tolerant control method of electronically controlled air suspension.

The article is organized as follows. In the next section, the model of the height adjustment system of the electronically controlled air suspension is established. After analyzing the fault mode by fault tree method, the controlled object and actuator model are established. In Section 3. The fault diagnosis mechanism based on the EKF group is designed and verified by simulation. In Section 4, active fault-tolerant control is designed based on the previous section. In Section 5, a hardware-in-the-loop simulation test bed is built to verify the fault diagnosis and active fault-tolerant control system. Finally, conclusions are drawn in Section 6.

## 2. Modeling and Failure Analysis of ECAS System

### 2.1. Modeling of Vehicle Height Adjustment System in ECAS System

To facilitate the design of the fault diagnosis filter, the vehicle ECAS system is simplified, and the corresponding vehicle height adjustment model of the vehicle ECAS system is established. This provides a basic platform for the implementation of actuator fault diagnosis. The vehicle ECAS dynamics model is established based on the following assumptions: (1) The sprung mass is a rigid body, and only vertical, pitch, and roll motions are considered; (2) Tyre damping is ignored; (3) The speed characteristic of the shock absorber is linear; (4) The ECAS system does not collide with the buffer block during the working process; (5) The rigidity of the frame and the body is sufficiently large, regardless of the vibration modes caused by the elasticity of the frame. Therefore, the ECAS dynamics model of the whole vehicle as shown in Figure 2 is established.

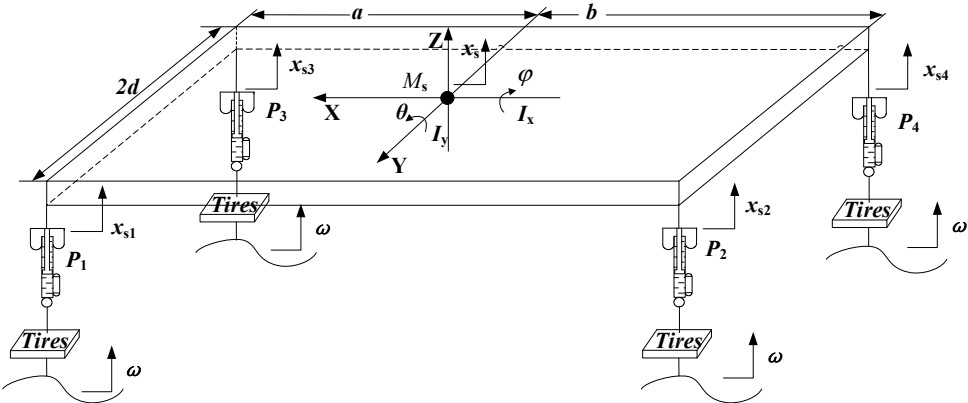

**Figure 2.** Dynamic model of the whole vehicle based on ECAS.

The ECAS vehicle dynamics equation is built according to the vehicle dynamics characteristics and related geometric relationships. Among them, the vehicle body vertical, pitch, and roll motion equations are as follows:

$$\begin{cases} M_s \ddot{x}_s = F_1 + F_2 + F_3 + F_4 \\ I_y \ddot{\theta} = b(F_2 + F_4) - a(F_1 + F_3) \\ I_x \ddot{\varphi} = (F_1 + F_2 - F_3 - F_4)d \end{cases} \tag{1}$$

where $M_s$ is the sprung mass of the whole vehicle, kg; $\ddot{x}_s$ is the vertical acceleration of the body center of mass; $F_1$, $F_2$, $F_3$, and $F_4$ are the forces on the front left, rear left, front right, and rear right body, respectively, N; $I_y$ is the moment of inertia of the car body around the Y-axis, km·m$^2$; $\ddot{\theta}$ is the pitch angular acceleration, rad/s$^2$; a and b are the distance from the center of mass of the car body to the front and rear axles, m; $I_x$ is the moment of inertia of the car body around the X-axis, km·m$^2$; $\ddot{\varphi}$ is the roll angular acceleration, rad/s$^2$; and $d$ is the 1/2 tread, m.

After establishing the vehicle dynamics model, a single-group air spring model is built. Assuming that the heat exchange during the gas flow is negligible, the charging and discharging process of the air spring can be regarded as a variable-volume adiabatic process. According to the first law of thermodynamics, when the solenoid valve is opened, the air spring opening inflates and discharges the variable mass model as follows:

$$\kappa R T \frac{dm}{dt} = \kappa P \frac{dv}{dt} + V \frac{dP}{dt} \tag{2}$$

When the solenoid valve is closed, $\frac{dm}{dt} = 0$, then Equation (4) can be rewritten as:

$$\frac{dP}{dt} = -\frac{\kappa P dV}{V dt} \tag{3}$$

Research has shown that when the height of the diaphragm air spring changes near the working position, its effective area and volume change rate can be regarded as fixed values. Therefore, the volume change of the air spring can be approximated as the spring vertical displacement change under the volume change rate, namely:

$$V = V_0 + \Delta V(x_{s1} - x_{u1}) \tag{4}$$

where $V_0$ is the initial volume of the air spring, m$^3$; and $\Delta V$ is the rate of change of air spring volume, m$^3$/m.



Combining Equations (2)–(4) can derive a complete air spring charging and discharging process model as follow:

$$V\dot{P} = -\kappa P \Delta V(\dot{x}_{s1} - \dot{x}_{u1}) + \kappa RT q_m \tag{5}$$

where $V$ is the air spring volume, m$^3$; $k$ is the adiabatic coefficient of air; $R$ is the gas constant, N·m/(kg·K); $T$ is the internal temperature of the air spring, °C; and $q_m$ is the gas mass flow rate when charging and discharging the air spring, kg/s.

According to the established vehicle ECAS dynamics model and the air spring charging and discharging process model, the mathematical model of the vehicle height adjustment of the vehicle ECAS system can be derived. Since the focus of the ECAS system model is the charging and discharging process of the air spring, the unsprung mass vibration and road input are combined as system noise. Then, the mathematical expression of the vehicle height adjustment of the complete vehicle ECAS system can be finally simplified as:

$$\begin{cases} M_s \ddot{x}_s = F_1 + F_2 + F_3 + F_4 \\ I_x \ddot{\varphi} = (F_1 + F_2 - F_3 - F_4)d \\ I_y \ddot{\theta} = b(F_2 + F_4) - a(F_1 + F_3) \\ V_1 \dot{P}_1 = -\kappa \Delta V P_1 \dot{x}_{s1} + \kappa RT q_{m1} \\ V_2 \dot{P}_2 = -\kappa \Delta V P_2 \dot{x}_{s2} + \kappa RT q_{m2} \\ V_3 \dot{P}_3 = -\kappa \Delta V P_3 \dot{x}_{s3} + \kappa RT q_{m3} \\ V_4 \dot{P}_4 = -\kappa \Delta V P_4 \dot{x}_{s4} + \kappa RT q_{m4} \end{cases} \tag{6}$$

where

$$\begin{cases} F_1 = (P_1 - P_a)A_e - m_{s1}g - C_1\dot{x}_{s1} - k_1\omega_1 \\ F_2 = (P_2 - P_a)A_e - m_{s2}g - C_2\dot{x}_{s2} - k_2\omega_2 \\ F_3 = (P_3 - P_a)A_e - m_{s3}g - C_3\dot{x}_{s3} - k_3\omega_3 \\ F_4 = (P_4 - P_a)A_e - m_{s4}g - C_4\dot{x}_{s4} - k_4\omega_4 \end{cases}, \begin{cases} V_1 x_{s1} \\ V_2 = V_{20} + \Delta V x_{s2} \\ V_3 = V_{30} + \Delta V x_{s3} \\ V_4 = V_{40} + \Delta V x_{s4} \end{cases}, \begin{cases} x_{s1} = x_s - a\theta + d\varphi \\ x_{s2} = x_s + b\theta + d\varphi \\ x_{s3} = x_s - a\theta - d\varphi \\ x_{s4} = x_s + b\theta - d\varphi \end{cases}$$

### 2.2. Actuator Failure Analysis and Modeling

Through the analysis of the potential failure modes of the components of the system, the reasons for the failures and their effects are drawn. The corresponding detection method is designed to greatly improve the reliability of the system.

To analyze the failure mode of the system, a fault tree analysis method is introduced here. The fault tree analysis method belongs to the method of graphical deduction. Through top-down or bottom-up logical deductive reasoning, the reasons leading to system failure are analyzed and expressed in a tree diagram.

In the fault tree analysis, the causal relationship between the faults is represented by event symbols, logic gate symbols, etc. Table 1 shows the basic symbols and their meanings used in the fault tree analysis method.

Figure 3 shows the air spring solenoid valve fault tree. It can be seen from the fault tree that the failure of the air spring solenoid valve of the ECAS system is mainly caused by the failure of the front left, front right, rear left, and rear right air spring solenoid valves. The main reasons for the failure of the above four solenoid valves include short circuits, open circuits, plugging of the valve core, spring fatigue, and leakage of the valve core.

**Table 1.** Basic symbols and meanings of fault tree.

| Event | Symbol | Description |
|---|---|---|
| Basic event | | The lowest level event that does not need to be ascertained |
| No extended Events | | Events that have little impact on the top event or whose cause cannot be known temporarily |
| Result Event | | Contains top and middle events that are always at the outputs |
| Transfer symbol | | This event indicates information transfer and avoids drawing repetition |
| Logic symbol: and gate | | Output events only occur when all input events occur |
| Logical symbols: or doors | | As long as one of the input events occurs, the output event occurs |

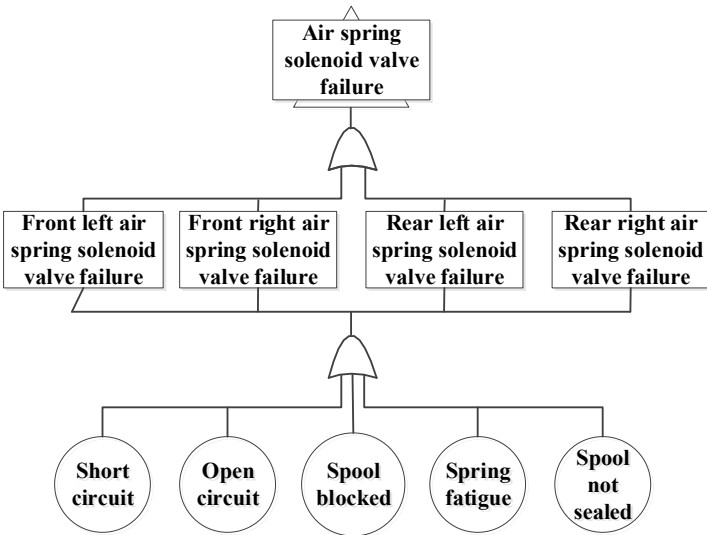

**Figure 3.** Fault tree of solenoid valves for air spring system.

After the failure mode analysis of the vehicle ECAS system, mathematical modeling is mainly carried out for the different failures of the actuator, namely the air spring solenoid valve.

The solenoid valve of the air spring is a high-speed switching solenoid valve, using the pulse-width modulation (PWM) control method. The relationship between the duty cycle and the flow rate is:

$$q_m = C_d \cdot A \cdot D \cdot \sqrt{\frac{2\Delta P}{\rho}} \tag{7}$$

where $C_d$ is the flow coefficient; $A$ is the flow area of the valve port, $m^2$; $D$ is the duty cycle; $\Delta P$ is the difference between input pressure and output pressure, $P_a$; and $\rho$ is the gas density, $kg/m^3$.

There are two common failure modes of solenoid valves. One is that the valve core is stuck due to the open circuit of the solenoid valve and cannot be opened normally. The flow through the solenoid valve becomes zero. Another failure is caused by the increase of internal friction of the solenoid valve. The core cannot reach the maximum displacement; that is, the valve port is not fully opened, resulting in a gain loss in the flow through the solenoid valve. Therefore, from the flow point of view, the fault behavior of the ECAS

system actuator is defined as stuck and constant gain. In the case of failure, the flow area of the valve port is expressed as follows:

$$A_f = n \cdot A + \beta \tag{8}$$

where $n$ is the fault gain factor, and $\beta$ is the dead value for failure.

Then, the stuck fault of the ECAS system actuator can be defined as $n = \beta = 0$, and the constant gain fault is $n \neq 0$ and $\beta \neq 0$.

According to Equations (7) and (8), when the actuator fails the control input of the ECAS system $u_f$ is as follow:

$$u_f = q_m = n \cdot u_i + \delta = C_d \cdot n \cdot A \cdot D \cdot \sqrt{\frac{2\Delta P}{\rho}} + C_d \cdot \beta \cdot D \cdot \sqrt{\frac{2\Delta P}{\rho}} \tag{9}$$

where $\delta$ is the input fault stuck value for the control, subscript $i = 1 - 4$.

The fault vector can be defined as:

$$F = \begin{bmatrix} f_1 \\ f_2 \\ f_3 \\ f_4 \end{bmatrix} = \begin{bmatrix} (n_1 - 1)u_1 + \delta_1 \\ (n_2 - 1)u_2 + \delta_2 \\ (n_3 - 1)u_3 + \delta_3 \\ (n_4 - 1)u_4 + \delta_4 \end{bmatrix} \tag{10}$$

Fault control input $U_f$ can be expressed as:

$$U_f = \begin{bmatrix} U_{f1} \\ U_{f2} \\ U_{f3} \\ U_{f4} \end{bmatrix} = \begin{bmatrix} u_1 \\ u_2 \\ u_3 \\ u_4 \end{bmatrix} + \begin{bmatrix} (n_1 - 1)u_1 + \delta_1 \\ (n_2 - 1)u_2 + \delta_2 \\ (n_3 - 1)u_3 + \delta_3 \\ (n_4 - 1)u_4 + \delta_4 \end{bmatrix} = U + F \tag{11}$$

## 3. Fault Diagnosis of ECAS System Based on Adaptive Threshold

### 3.1. Fault Diagnosis System Architecture Based on KEFs

When the actuator of the ECAS system fails, the controller may adjust the charging and bleeding process of the air spring abnormally. Therefore, the actuator fault detection and isolation strategy shown in Figure 4 is designed for the vehicle ECAS system.

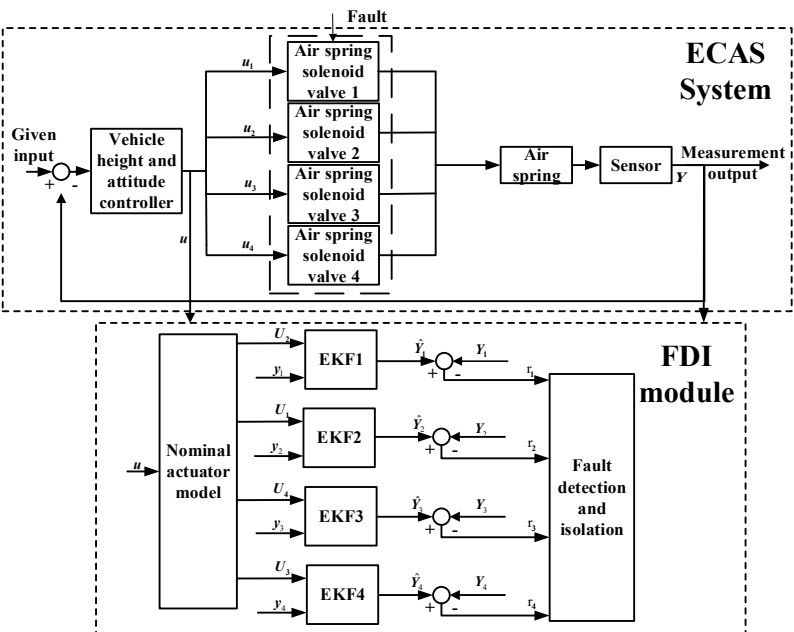

**Figure 4.** Actuator fault detection & isolation strategy of the vehicle ECAS system.

The extended Kalman filter (EKF) algorithm is widely used for state estimation in nonlinear systems. The main idea is to use Taylor's formula to transform the nonlinear model into a linear model and then perform Kalman filtering. Therefore, in view of the nonlinear characteristics of the ECAS system, the extended Kalman filter algorithm is used to design the fault diagnosis filter.

To design EKF1, selecting the state variable $X_1 = \left[\theta\dot{\theta}\varphi\ \dot{\varphi}x_s\dot{x}_sx_2\dot{x}_2P_2\right]^T$, defining measurement output $Y_1 = \left[x_2\ddot{x}_2P_2\right]^T$, control input $U_2 = [qm_2]^T$; For EKF2, selecting state variable $X_2 = \left[\theta\dot{\theta}\varphi\ \dot{\varphi}x_s\dot{x}_sx_1\dot{x}_1P_1\right]^T$, defining measurement output $Y_2 = \left[x_1\ddot{x}_1P_1\right]^T$, control input $U_1 = [qm_1]^T$; For EKF3, selecting state variable $X_3 = \left[\theta\dot{\theta}\varphi\ \dot{\varphi}x_s\dot{x}_sx_4\dot{x}_4P_4\right]^T$, defining measurement output $Y_3 = \left[x_4\ddot{x}_4P_4\right]^T$, control input $U_4 = [qm_4]^T$; For EKF4, selecting state variable $X_4 = \left[\theta\dot{\theta}\varphi\ \dot{\varphi}x_s\dot{x}_sx_3\dot{x}_3P_3\right]^T$, defining measurement output $Y_4 = \left[x_3\ddot{x}_3P_3\right]^T$, and control input $U_3 = [qm_3]^T$.

According to Equation (5) and the selected state variables, measurement output, and control input, write the corresponding system and measurement equations, respectively. The general form of the system and measurement equation is as follows:

$$\begin{cases} \dot{X} = f(X) + g(X)U + q(X)\omega \\ \qquad\qquad Y = h(X) + v \end{cases} \tag{12}$$

$f(X)$, $g(X)$, $q(X)$ and $h(X)$ derived from EKF1, EKF2, EKF3, and EKF4, respectively.
According to the Equation (11) design filter equation, the general form of the equation is:

$$\begin{cases} \dot{\hat{X}} = f(\hat{X}) + g(\hat{X})U + L_k(Y - \hat{Y}) \\ \qquad\qquad \hat{Y} = h(\hat{X}) \end{cases} \tag{13}$$

where $\hat{X}$ is the estimate for state variables, $\hat{Y}$ is the estimate for measurement output, $Y$ is the output for measurement, $U$ is the input matrix for control, and $L_k$ is the filter gain matrix. Process noise $\omega$ and measurement noise $v$ are mutually uncorrelated Gaussian white noise. The probability distribution characteristics are as follows:

$$\begin{cases} E(\omega_k) = 0,\ Cov(\omega_k,\ \omega_j) = Q_k\delta_{kj} \\ E(v_k) = 0,\ Cov(v_k,\ v_j) = R_k\delta_{kj} \\ \qquad\qquad Cov(\omega_k,\ v_j) = 0 \end{cases} \tag{14}$$

where $Q_k$ is the process noise covariance matrix, and $R_k$ is the measurement noise covariance matrix. The EKF algorithm is shown in Figure 5, where $F_k = \frac{\partial f(x_k)}{\partial x_k}|x_k = \hat{x}_k$, $H_k = \frac{\partial h(x_k)}{\partial x_k}|x_k = \hat{x}_k$, $\Gamma_k = \frac{\partial f(x_k)}{\partial \omega_k}|x_k = \hat{x}_k$, $\hat{z}_k = h(\hat{x}_{k,\ k-1})$.

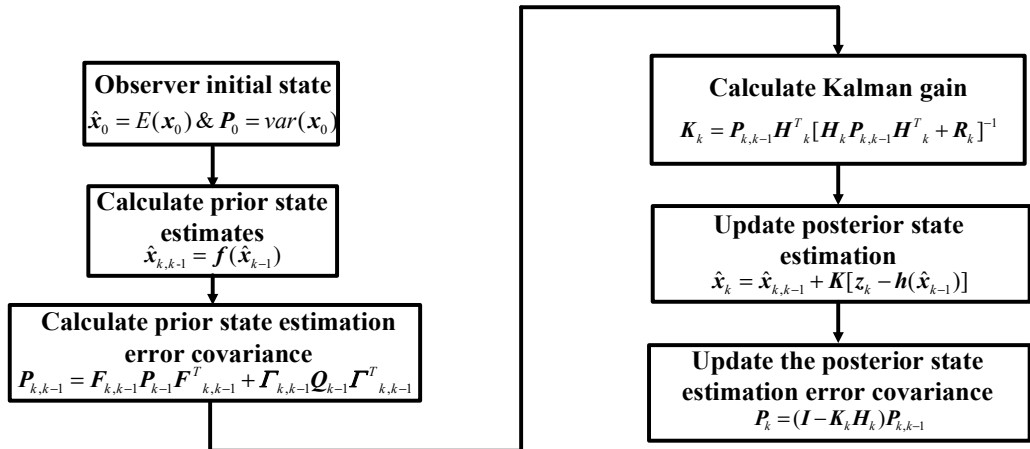

**Figure 5.** EKF algorithm flow.

### *3.2. Calculation of Adaptive Threshold*

Threshold selection is an important step to achieve fault detection and isolation. The adaptive threshold changes with system input, which can effectively reduce the probability of missed diagnosis and misdiagnosis. In this section, an adaptive threshold is designed according to the system model error and system input, so as to minimize the probability of missed diagnosis and misdiagnosis. The system model error includes linearization error and parameter uncertainty error. Firstly, the system equation of EKF1 is taken as an example to illustrate the analysis of the linearization error and parameter uncertainty error

1.    Linearization error

The ECAS system equation has nonlinear characteristics. In the design process of the extended Kalman filter, the system equations need to be linearized, resulting in linearization errors. Equation (12) can be written in the form of $\dot{X} = AX + BU$ $X$. Thus, the state transition matrix $A$ and the control input coefficient matrix $B$ are obtained.

$$
A = \begin{bmatrix}
0 & 1 & 0 & 0 & 0 & 0 & 0 & 0 & 0 \\
0 & 0 & 0 & 0 & 0 & 0 & 0 & \frac{-bC_2}{I_y} & \frac{bA_e}{I_y} \\
0 & 0 & 0 & 1 & 0 & 0 & 0 & 0 & 0 \\
0 & 0 & 0 & 0 & 0 & 0 & 0 & \frac{-C_2 d}{I_x} & \frac{dA_e}{I_x} \\
0 & 0 & 0 & 0 & 0 & 1 & 0 & 0 & 0 \\
0 & 0 & 0 & 0 & 0 & 0 & 0 & \frac{-C_2}{m_s} & \frac{A_e}{m_s} \\
0 & 0 & 0 & 0 & 0 & 0 & 0 & 1 & 0 \\
0 & 0 & 0 & 0 & 0 & 0 & 0 & \frac{-C_2}{m_{s2}} & \frac{A_e}{m_{s2}} \\
0 & 0 & 0 & 0 & 0 & 0 & 0 & \frac{-\kappa \Delta \dot{V} P_0}{V_{20}} & 0
\end{bmatrix} \tag{15}
$$

$$
B = \begin{bmatrix} 0 & 0 & 0 & 0 & 0 & 0 & 0 & 0 & \frac{\kappa RT}{V_{20}} \end{bmatrix}^T \tag{16}
$$

Thus, the linearization error matrix can be obtained as:

$$
\Delta A_1 = \begin{bmatrix}
0 & 0 & 0 & 0 & 0 & 0 & 0 & 0 & 0 \\
0 & 0 & 0 & 0 & 0 & 0 & 0 & 0 & 0 \\
0 & 0 & 0 & 0 & 0 & 0 & 0 & 0 & 0 \\
0 & 0 & 0 & 0 & 0 & 0 & 0 & 0 & 0 \\
0 & 0 & 0 & 0 & 0 & 0 & 0 & 0 & 0 \\
0 & 0 & 0 & 0 & 0 & 0 & 0 & 0 & 0 \\
0 & 0 & 0 & 0 & 0 & 0 & 0 & 0 & 0 \\
0 & 0 & 0 & 0 & 0 & 0 & 0 & 0 & 0 \\
0 & 0 & 0 & 0 & 0 & 0 & 0 & \Delta A_{11} & 0
\end{bmatrix} \tag{17}
$$

$$\Delta B_1 = \begin{bmatrix} 0 & 0 & 0 & 0 & 0 & 0 & 0 & 0 & \Delta B_{11} \end{bmatrix}^T \tag{18}$$

where $\Delta A_{11} = \frac{-\kappa \Delta V P_2}{V_{20} + \Delta V x(7)} + \frac{\kappa \Delta V P_0}{V_{20}}$, and $\Delta B_{11} = \frac{\kappa RT}{V_{20}} - \frac{\kappa RT}{V_{20} + \Delta V x(7)}$.

2.  Parameter uncertainty error

In the ECAS vehicle model, the uncertain parameters mainly include the sprung mass and the damping of the shock absorber. The sprung mass has the characteristics of uneven distribution and changes with vehicle masses. The damping value of the shock absorber is also different under different working temperatures. Therefore, the parameter uncertainty is introduced as follows:

$$\begin{cases} \Delta m_{s1} = m_{s1} - m_{s1}{}^{real} \\ \Delta m_{s2} = m_{s2} - m_{s2}{}^{real} \\ \Delta m_{s3} = m_{s3} - m_{s3}{}^{real} \\ \Delta m_{s4} = m_{s4} - m_{s4}{}^{real} \\ \Delta m_s = m_s - m_s{}^{real} \end{cases} \tag{19}$$

$$\begin{cases} \Delta C_1 = C_1 - C_1{}^{real} \\ \Delta C_2 = C_2 - C_2{}^{real} \\ \Delta C_3 = C_3 - C_3{}^{real} \\ \Delta C_4 = C_4 - C_4{}^{real} \end{cases} \tag{20}$$

The parameter value with real superscript represents the actual parameter value or the floating limit of the parameter. From this, the parameter uncertainty error matrix is derived as follows:

$$\Delta A_2 = \begin{bmatrix} 0 & 0 & 0 & 0 & 0 & 0 & 0 & 0 & 0 \\ 0 & 0 & 0 & 0 & 0 & 0 & 0 & \Delta A_{21} & 0 \\ 0 & 0 & 0 & 0 & 0 & 0 & 0 & 0 & 0 \\ 0 & 0 & 0 & 0 & 0 & 0 & 0 & \Delta A_{22} & 0 \\ 0 & 0 & 0 & 0 & 0 & 0 & 0 & 0 & 0 \\ 0 & 0 & 0 & 0 & 0 & 0 & 0 & \Delta A_{23} & \Delta A_{24} \\ 0 & 0 & 0 & 0 & 0 & 0 & 0 & 0 & 0 \\ 0 & 0 & 0 & 0 & 0 & 0 & 0 & \Delta A_{25} & \Delta A_{26} \\ 0 & 0 & 0 & 0 & 0 & 0 & 0 & 0 & 0 \end{bmatrix} \tag{21}$$

$$\Delta B_2 = \begin{bmatrix} 0 & 0 & 0 & 0 & 0 & 0 & 0 & 0 & 0 \end{bmatrix}^T \tag{22}$$

where $\Delta A_{21} = \frac{-bC_2}{I_y} + \frac{bC_2{}^{real}}{I_y}$, $\Delta A_{22} = \frac{-C_2 d}{I_x} + \frac{C_2{}^{real} d}{I_x}$, $\Delta A_{23} = \frac{-C_2}{m_s} + \frac{C_2{}^{real}}{m_s{}^{real}}$, $\Delta A_{24} = \frac{A_e}{m_s} - \frac{A_e}{m_s{}^{real}}$, $\Delta A_{25} = \frac{-C_2}{m_{s2}} + \frac{C_2{}^{real}}{m_{s2}{}^{real}}$, and $\Delta A_{26} = \frac{A_e}{m_{s2}} - \frac{A_e}{m_{s2}{}^{real}}$

The model error matrix is introduced, and the system input is considered to determine the adaptive threshold. Then, the system can be expressed to:

$$\dot{X} = (A + \Delta A)X + (B + \Delta B)U \tag{23}$$

where $\Delta A = \Delta A_1 + \Delta A_2$, and $\Delta B = \Delta B_1 + \Delta B_2$.

Model error can be defined as $\varepsilon = X - \hat{X}$. Equations (13) and (23) are combined, so that the model error can be expressed as follow:

$$\dot{\varepsilon} = (A + L_k C)\varepsilon + (\Delta A_1 + \Delta A_2)X + (\Delta B_1 + \Delta B_2)U \tag{24}$$

Integrating Equation (24), the error $\varepsilon$ can be deduced as:

$$\begin{aligned} \varepsilon = \quad & \varepsilon^{(A+L_k C)t}\varepsilon(0) + \int_0^t \varepsilon^{(A+L_k C)(t-\tau)}(\Delta A_1 + \Delta A_2)X(\tau)d\tau \\ & + \int_0^t \varepsilon^{(A+L_k C)(t-\tau)}(\Delta B_1 + \Delta B_2)U(\tau)d\tau \end{aligned} \tag{25}$$

The adaptive threshold for fault detection is given from the above equation as:

$$h = \varepsilon + c \tag{26}$$

where $c$ calculates the acceptable deviation.

### 3.3. Fault Detection

The output residual is defined according to fault detection and isolation strategy as $r = Y - \hat{Y}$. For the vehicle ECAS system, the residual characteristic description shown in Table 2 can be obtained.

**Table 2.** Characterization of residuals.

| Residual | $r_1^{(1)}$ | $r_2^{(1)}$ | $r_3^{(1)}$ | $r_4^{(1)}$ | $r_1^{(2)}$ | $r_2^{(2)}$ | $r_3^{(2)}$ | $r_4^{(2)}$ | $r_1^{(3)}$ | $r_2^{(3)}$ | $r_3^{(3)}$ | $r_4^{(3)}$ |
|---|---|---|---|---|---|---|---|---|---|---|---|---|
| No Failure | 0 | 0 | 0 | 0 | 0 | 0 | 0 | 0 | 0 | 0 | 0 | 0 |
| Actuator 1 failure | 0 | 1 | 0 | 0 | 0 | 1 | 0 | 0 | 0 | 1 | 0 | 0 |
| Actuator 2 failure | 1 | 0 | 0 | 0 | 1 | 0 | 0 | 0 | 1 | 0 | 0 | 0 |
| Actuator 3 failure | 0 | 0 | 0 | 1 | 0 | 0 | 0 | 1 | 0 | 0 | 0 | 1 |
| Actuator 4 failure | 0 | 0 | 1 | 0 | 0 | 0 | 1 | 0 | 0 | 0 | 1 | 0 |

In Table 2, $r_i^{(j)}$ is the output estimated residual, $i$ is the extended Kalman filter number ($i = 1 - 4$), and $j$ is the measurement output number. $j = 1, 2, 3$ represent the height change of the air spring, the vertical acceleration at the four corners of the body, and the internal air pressure of the air spring, respectively. Actuators 1–4 represent the front left, rear left, front right, and rear right air spring solenoid valves, respectively. Taking the residual $r$ as the fault detection indicator, there are three fault detection indicators, including the air spring height estimation residual $r_i^{(1)}$ and the vertical acceleration estimation residual error at the four corners of the car body $r_i^{(2)}$, and the air spring pressure estimation residual $r_i^{(3)}$. Each extended Kalman filter will produce the above three fault detection indicators (that is, $r_i^{(1)}$, $r_i^{(2)}$ and $r_i^{(3)}$). Comparing the fault detection index with the adaptive threshold h, it can be detected whether the actuator has a fault.

$$\begin{cases} index \geq h, \text{Failure} \\ index < h, \text{Nofailure} \end{cases} \tag{27}$$

According to Equation (27), when the fault detection index value is greater than or equal to the detection threshold, the actuator has failed. The corresponding detection index $r_i^{(j)} = 1$; when the detection index value is less than the detection threshold, the actuator has not failed. The corresponding detection index $r_i^{(j)}$ is equal to 0.

As long as one of the three fault detection indicators exceeds the threshold, it is considered that a fault has occurred. The advantage of setting three fault detection indicators is to further reduce the missed diagnosis rate and increase the reliability and effectiveness of fault detection. By looking up Table 2, you can not only know whether the actuator is malfunctioning, but also determine the location of the malfunctioning actuator; that is, fault detection and isolation are realized.

### 3.4. Simulation and Analysis

To verify the proposed fault diagnosis program, four fault behaviors shown in Table 3 are selected. Fault 1 is a stuck fault, and Faults 2–4 are constant gain faults. In the fault behavior, $n$ represents the gain coefficient, and $\delta$ represents the fault stuck value. The flow area of the valve port can be calculated from Equation (8). Therefore, the fault behavior represents the flow area of the valve port, corresponding to four fault behaviors.

Actuators 1–4, respectively, represent the front left, rear left, front right, and rear right air spring solenoid valves. The fault diagnosis module was started at the same time as the vehicle height adjustment was started at 5 s, and the fault occurred at 8 s. The first 5 s is the process of the air spring model in AMEsim gradually returning to a steady state. The simulation results of Faults 1 and 2 detections are shown in Figures 6 and 7.

**Table 3.** Description of fault behavior.

| Fault Number | | 1 | 2 | 3 | 4 |
|---|---|---|---|---|---|
| Moment of failure/s | | | | 8 | |
| Fault behavior | $n$ | 0 | 0.2 | 0.4 | 0.6 |
| | $\delta$ | 0 | 0 | 0 | 0 |

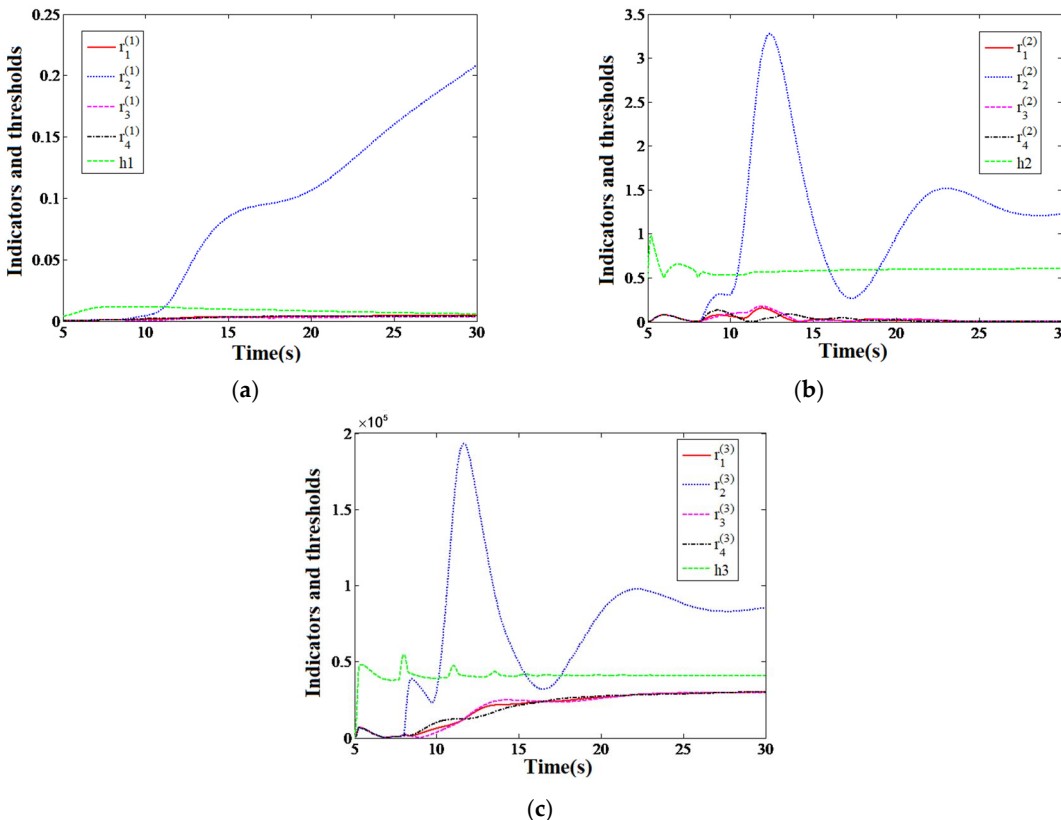

**(a)**

**(b)**

**(c)**

**Figure 6.** Changes of fault detection indicators under Fault 1: (**a**) Fault detection index $r_i^{(1)}$ and threshold h1; (**b**) Fault detection index $r_i^{(2)}$ and threshold h2; (**c**) Fault detection index $r_i^{(3)}$ and threshold h3.

As shown in Figure 6, before the occurrence of Fault 1, the estimated residuals of the three fault detection indicators, namely displacement, acceleration, and air pressure output, are smaller than the adaptive threshold. The fault occurs at the 8th second, and the residual error $r_2^{(1)}$, $r_2^{(2)}$ and $r_2^{(3)}$ output by EKF2 all exceeds the adaptive threshold. The residual outputs by other filters still fluctuate around zero or are less than the adaptive threshold. According to Figure 7 and the residual characteristics in Table 2, the front left air spring solenoid valve is malfunctioning. The fault detection time is 11.1 s, 10.4 s, and 10.1 s, respectively. If one of $r_2^{(1)}$, $r_2^{(2)}$ and $r_2^{(3)}$ exceeds the threshold, it is considered that a fault has occurred. Therefore, the actuator failure was detected at 10.1 s.

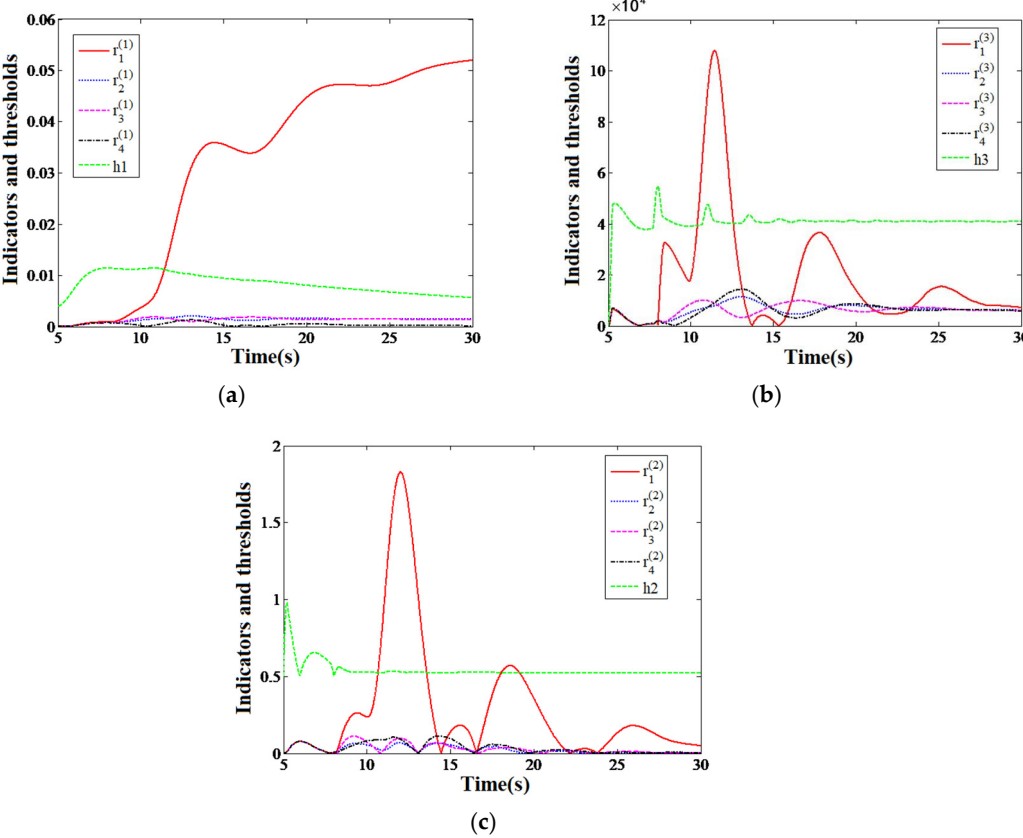

**Figure 7.** Change of fault detection index under Fault 2: (**a**) Fault detection index $r_i^{(1)}$ and threshold h1; (**b**) Fault detection index $r_i^{(2)}$ and threshold h2; (**c**) Fault detection index $r_i^{(3)}$ and threshold h3.

As shown in Figure 7, before Fault 2 occurs, the fault detection indicators are all less than the adaptive threshold. After the fault occurs, the output residuals of EKF1 $r_1^{(1)}$, $r_1^{(2)}$, $r_1^{(3)}$ all exceed the adaptive threshold. The residual outputs by other filters are still smaller than the adaptive threshold, and the fault detection time is 11.4 s, 10.6 s, and 10.4 s, respectively. Therefore, the actuator failure is detected at 10.4 s. According to Figure 7 and the residual characteristic Table 2, it can be judged that the rear left air spring solenoid valve is malfunctioning.

Similarly, simulation verification was performed for Failures 3 and 4. According to the simulation results and the residual characteristics in Table 2, it can be accurately known that the fault occurred in the front right and rear right air spring solenoid valves. In summary, the ECAS system fault diagnosis system based on the adaptive threshold is accurate and effective. It can correctly judge whether there is a fault and the location of the corresponding faulty solenoid valve.

## 4. Design and Simulation of Active Fault Tolerant Control

### 4.1. Design of Active Fault Tolerant Control

An active fault-tolerant control strategy is designed for the actuator failure of the vehicle ECAS system, shown in Figure 8. The active fault-tolerant control decision-making module (the content of this module is shown in Table 4) judges whether a fault has occurred according to the information sent by the fault detection and isolation module. If there is no fault, use the original controller. If a fault is detected, the fault estimation value sent by the fault estimation module is judged as whether it is stuck or a constant gain fault. If it is a stuck fault, the vehicle height adjustment is stopped immediately. If it is a constant gain fault, the estimated value $\hat{A}_1$ of the valve port area calculated from the estimated value of the fault is sent to the controller. The original valve port area parameter $A_0$ of the controller is replaced so that the duty ratio of the solenoid valve is increased, and the height change

rate of the air spring corresponding to the faulty solenoid valve is increased. This can improve the inconsistency of air spring changes at the four corners and, ultimately, increase the vehicle height adjustment speed, as well as improve the body posture.

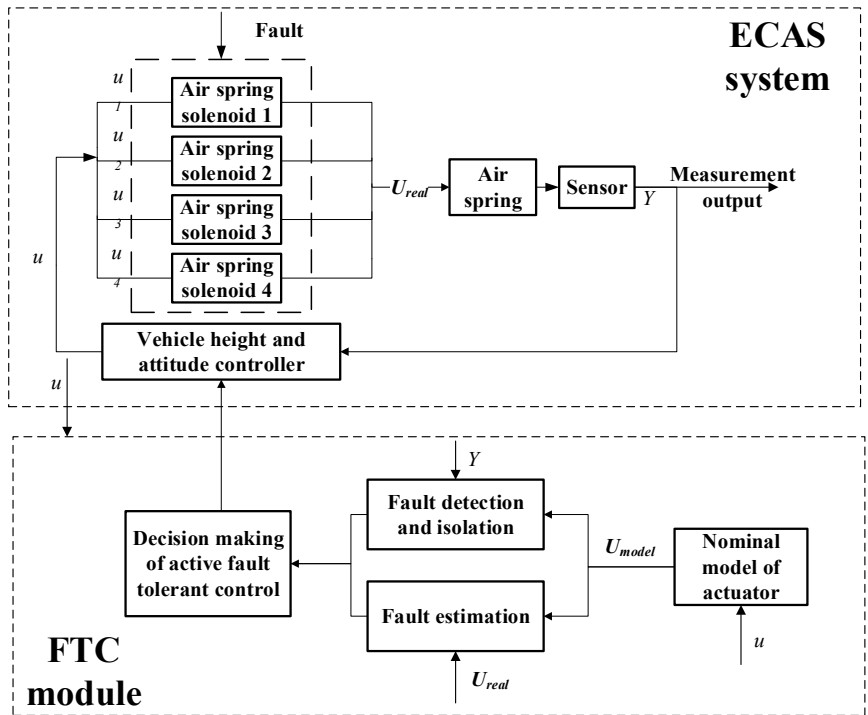

**Figure 8.** Active fault tolerant control strategy of vehicle ECA system.

**Table 4.** Contents of active fault tolerant control decision module.

| Fault Type | Fault Tolerance Measures |
|---|---|
| No fault | The original controller is adopted |
| Constant gain fault | Online adjustment of controller parameters |
| Stuck fault | Close all solenoid valves and stop height adjustment |

*4.2. Simulation of Active Fault Tolerant Control*

Taking Fault 1 (stuck) and Fault 2 (constant gain) shown in Table 3 as examples, the simulation results are as follows.

From Figure 9b,c, it can be seen that under normal circumstances, the pitch and roll angles of the vehicle body during the vehicle height adjustment process (5 s to 22.5 s) are well controlled. In the case of Fault 1, the height of the front left air spring stops increasing because the front left air spring solenoid valve is stuck. Its changes cannot be synchronized with other air springs. This causes the pitch angle and roll angle during the height adjustment process to rapidly increase to about 0.0055 rad and −0.0066 rad, respectively, and the body attitude deteriorates. Under fault-tolerant control, the vehicle height adjustment stops because all air spring solenoid valves are closed. Therefore, the pitch angle and roll angle of the body are stabilized at about 0.0005 rad and −0.0012 rad, to avoid further deterioration of the body attitude.

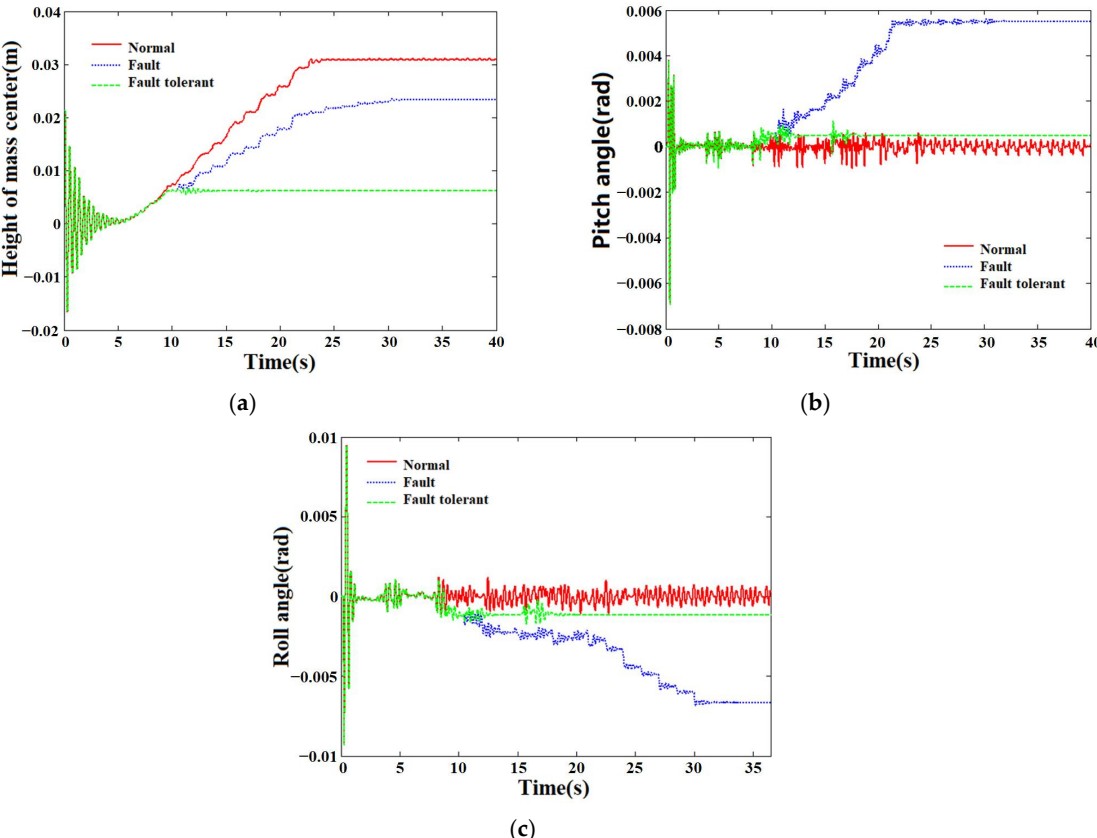

**Figure 9.** Performance comparison before and after fault tolerant control under Fault 1 (stuck): (**a**) Relative height of mass center; (**b**) Pitch angle; (**c**) Roll angle.

It can be seen from Figure 10 that when Fault 2 occurs, the vehicle height adjustment speed without fault-tolerant control decreases, and the pitch and roll angle peaks increase. This is due to the fact that the solenoid valve port of the rear left air spring cannot be fully opened, and a constant gain failure has occurred, resulting in a decrease in flow. After performing fault-tolerant control, the duty ratio of the rear left air spring solenoid valve is increased by adjusting the controller parameters online. After entering, the flow rate of the left air spring increases, and the vehicle height adjustment time decreases. Compared with fault-tolerant control, the vehicle height adjustment time is improved by about 15.3%. The peak pitch angle and roll angle are reduced, and the improvement in the peak pitch angle of the body is about 43.8%. The peak roll angle improvement rate is about 37.5%.

Faults 3 and 4 are also constant gain faults. Since the air spring solenoid valve cannot be fully opened, the flow into the solenoid valve is reduced. The simulation results show that the vehicle height adjustment speed decreases, and the pitch angle and roll angle peaks increase when there is no fault-tolerant control. After performing fault-tolerant control, the vehicle height adjustment time range, pitch angle, and roll angle peak are all improved.

In summary, when the actuator fails, the designed active fault-tolerant control method can effectively improve the vehicle height adjustment and attitude control performance under the fault compared to the case of no fault-tolerant control.

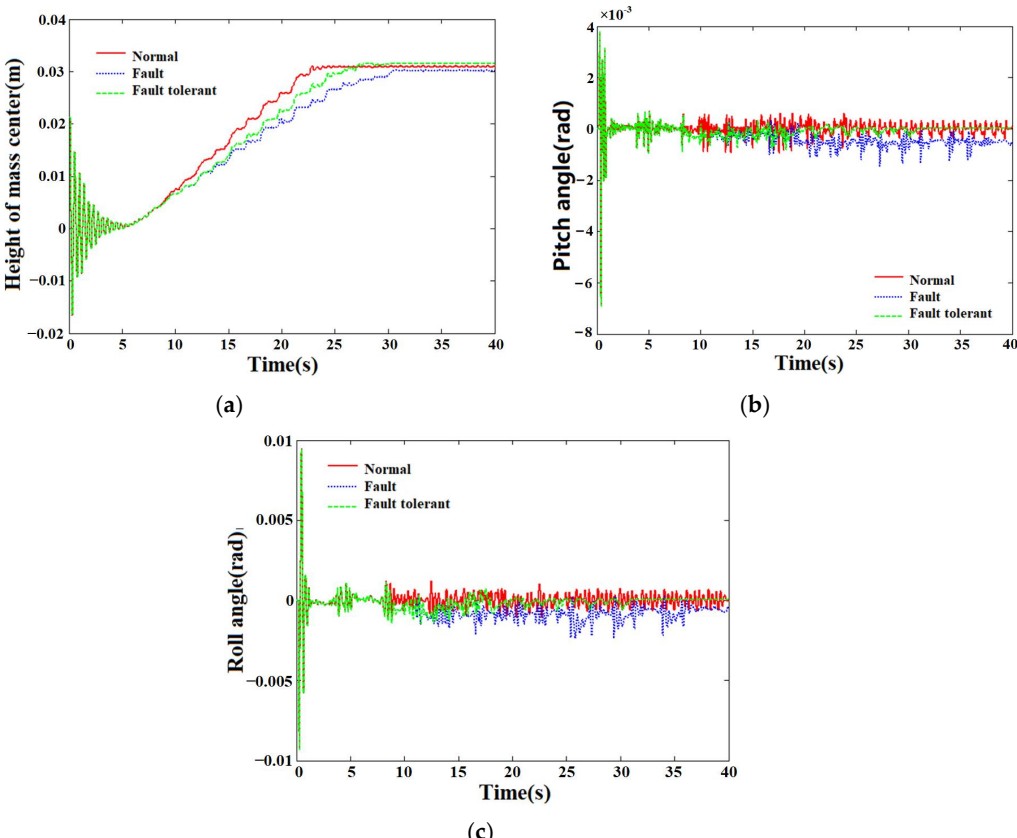

**Figure 10.** Performance comparison before and after fault tolerant control under Fault 2 (constant gain): (**a**) Relative height of mass center; (**b**) Pitch angle; (**c**) Roll angle.

## 5. Hardware In-Loop Simulation System

### 5.1. Hardware Platform

The fault-tolerant control HiL test platform of the vehicle ECAS system simulates the input and output signals through various boards. The connection between the real controller and the controlled object simulation model can be realized to form a complete loop. To verify the actual operation and feasibility of the fault-tolerant control algorithm model in the real controller, the hardware test is shown in Figure 11.

Under the existing hardware platform, building an HiL test system that is mainly divided into three steps includes establishing the controlled object model, developing the control model, and creating the system engineering file and user interface.

The controlled object model of the vehicle ECAS system is obtained by specifying relevant settings based on the AMESim model, including (1) online parameter setting, (2) observation variable setting, and (3) external interface setting. After applying the corresponding settings, you can compile and generate *.dll file, and load it into the system project file. The control model is built based on D2P rapid control prototyping technology, and NI VeriStand software is used to create project files and user interfaces.

NI VeriStand software is a software environment for configuring real-time test applications. Its functions include configuring the operating system, board, vehicle model, input/output interface with the actual controller, and user interface. NI VeriStand software carries out hardware-in-the-loop simulation by interacting with the signal of the actual controller. Researchers can monitor the simulation process in real time through the upper computer. The test principle of the HiL platform is shown in Figure 12.

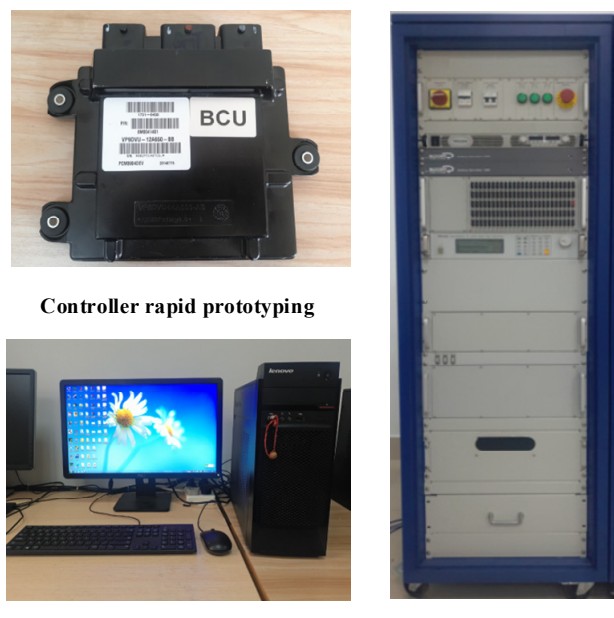

**Figure 11.** Hardware composition of fault-tolerant control HiL test platform for ECAS system.

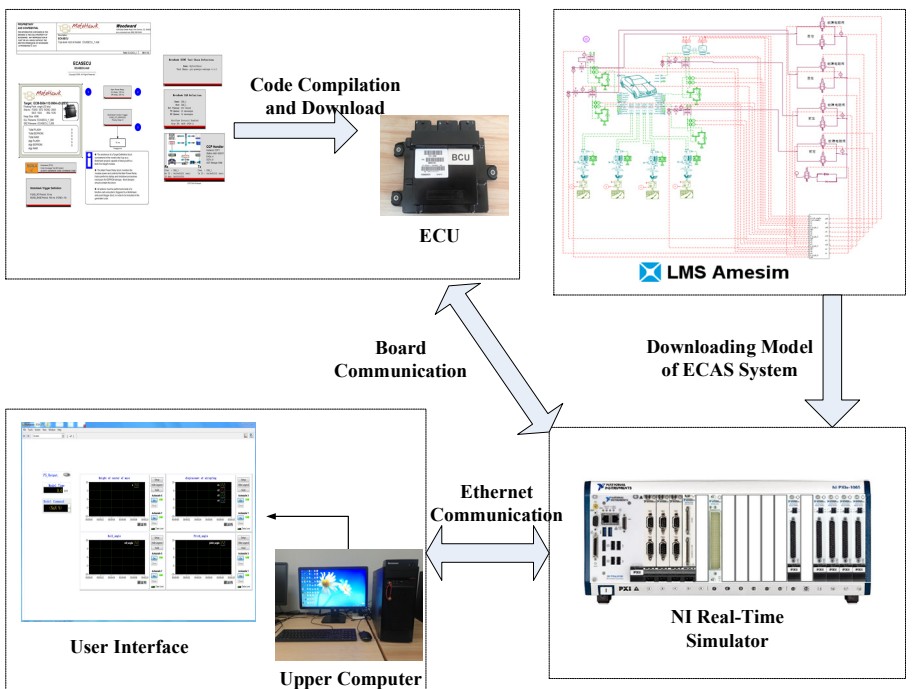

**Figure 12.** The working principle of HiL test platform.

*5.2. Results and Analysis*

Fault Behaviors 1 and 2 in Table 3 are selected for the fault-tolerant control HiL test, and the results are shown in Figures 13 and 14.

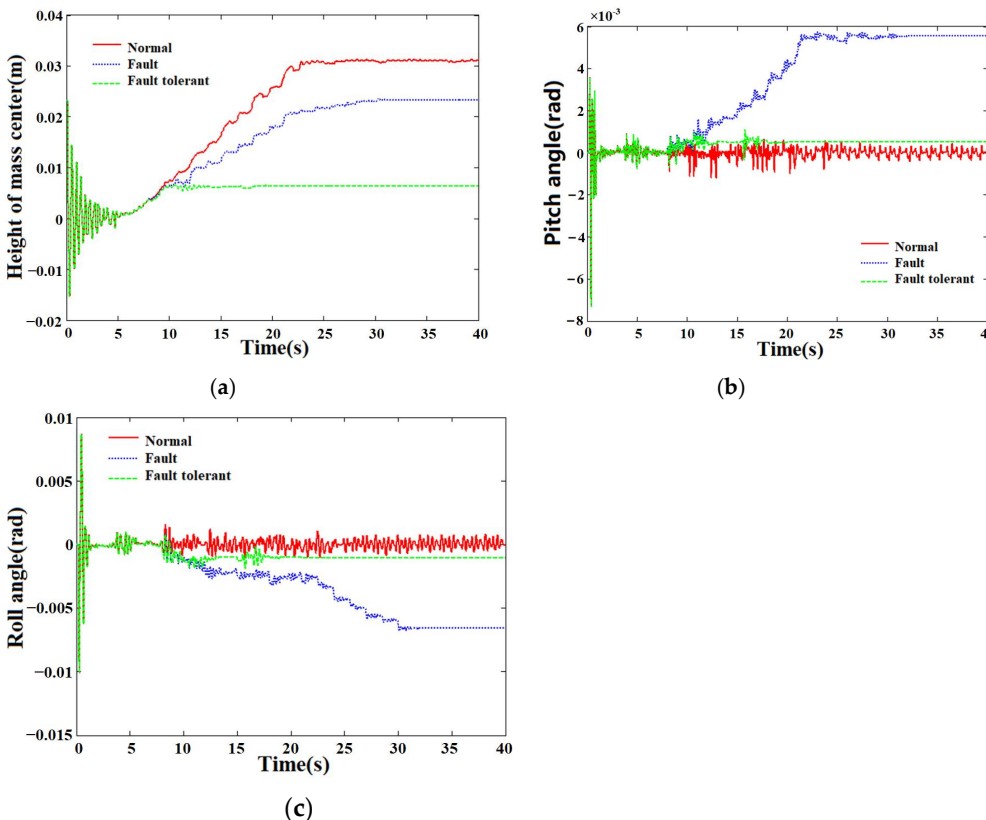

**Figure 13.** Performance comparison before and after fault tolerant control under Fault 1: (**a**) Relative height of mass center; (**b**) Pitch angle; (**c**) Roll angle.

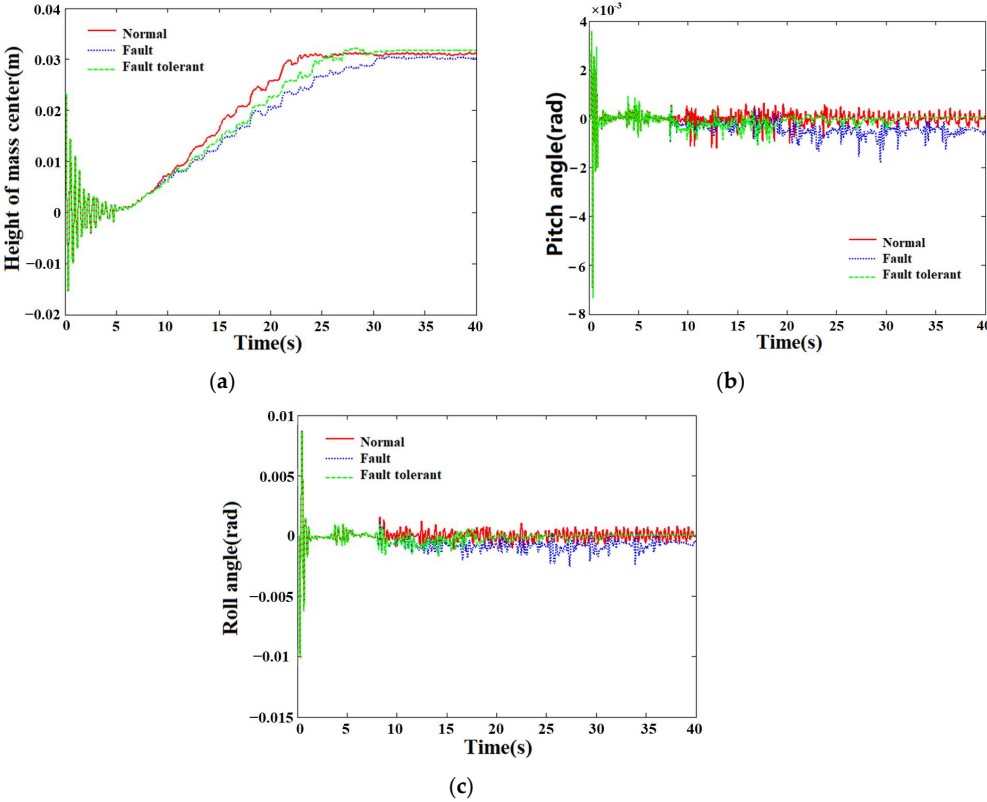

**Figure 14.** Performance comparison before and after fault tolerant control under Fault 2: (**a**) Relative height of mass center; (**b**) Pitch angle; (**c**) Roll angle.

As shown in Figure 13, after the vehicle height starts to increase (that is, after 5 s), it will reach the target height after about 17.5 s (that is, the 22.5 s) normally. After the current left air spring solenoid valve has a stuck fault, if there is no fault-tolerant control, the relative height of the body's center of mass will continue to increase to about 0.0232 m, and the pitch angle and roll angle will rapidly increase to about 0.0056 rad and −0.0066 rad, respectively. Under fault-tolerant control, about 2.1 s after the fault occurs, the air spring solenoid valve is closed, and the vehicle height adjustment stops. At this time, the pitch angle and roll angle of the body are stabilized at about 0.0005 rad and −0.0011 rad. Compared with the model-in-loop (MiL) simulation of the Fault 1 situation in Section 3.4, it can be found that the data results are similar. For Fault 2, the HiL test curve is shown in Figure 14.

The data results of the MiL and HiL simulation tests are given in Tables 5 and 6, respectively. By comparing Tables 5 and 6, it can be found that the HiL test data results of faults are basically consistent with the MiL simulation data results.

**Table 5.** Analysis of simulation results of Faults 2–4.

| Fault Number | Performance Index | Failure (No Fault Tolerance) | Fault Tolerant Control | Improvement Range |
|---|---|---|---|---|
| | Height adjustment time (s) | 25.5 | 21.6 | 15.3% |
| 2 | Peak pitch angle (RAD) | | 0.0009 | 43.8% |
| | Peak roll angle (RAD) | 0.0024 | 0.0015 | 37.5% |
| | Height adjustment time (s) | 24.2 | 20.4 | 15.7% |
| 3 | Peak pitch angle (RAD) | 0.0094 | 0.001 | 89.4% |
| | Peak roll angle (RAD) | 0.0063 | 0.0013 | 79.4% |
| | Height adjustment time (s) | 20.8 | 19.2 | 7.7% |
| 4 | Peak pitch angle (RAD) | 0.0027 | 0.0011 | 59.3% |
| | Peak roll angle (RAD) | 0.0017 | 0.0012 | 29.4% |

**Table 6.** HiL test result analysis of Faults 2–4.

| Fault Number | Performance Index | Failure (No Fault Tolerance) | Fault Tolerant Control | Improvement Range |
|---|---|---|---|---|
| | Height adjustment time (s) | 25.5 | 21.6 | 15.3% |
| 2 | Peak pitch angle (RAD) | 0.0018 | 0.001 | 44.4% |
| | Peak roll angle (RAD) | 0.0025 | 0.0016 | 36% |
| | Height adjustment time (s) | 24.2 | 20.4 | 15.7% |
| 3 | Peak pitch angle (RAD) | 0.0091 | 0.00098 | 89.2% |
| | Peak roll angle (RAD) | 0.0061 | 0.0016 | 73.8% |
| | Height adjustment time (s) | 20.8 | 19.2 | 7.7% |
| 4 | Peak pitch angle (RAD) | 0.003 | 0.0012 | 60% |
| | Peak roll angle (RAD) | 0.0017 | 0.0013 | 23.5% |

In summary, the HiL test results of fault diagnosis and fault-tolerant control strategy are basically consistent with the MiL simulation results in Section 3.4. It shows that the fault diagnosis and fault-tolerant control model designed in the actual controller can operate normally. It can effectively realize the fault detection isolation of the controlled object under the actuator failure and the improvement of the vehicle height adjustment and attitude control performance.

## 6. Conclusions

In this paper, the fault diagnosis and active fault-tolerant control of the ECAS system under actuator fault are studied. Based on the fault model, an extended Kalman filter bank with an adaptive threshold is designed for fault diagnosis. In addition, online adjustment is adopted for fault-tolerant control.

Firstly, the ECAS vehicle model is simplified, and the mathematical model of vehicle height regulation is established as the basis of fault-tolerant control. The faults are classified

into constant gain faults and stuck faults by the fault tree method. According to the fault type, an accurate fault mathematical model is established. Then, an adaptive threshold extended Kalman filter bank is designed as the observer. Each residual is compared with the adaptive threshold. Therefore, the fault location and type can be accurately located. This method improves the accuracy and speed of diagnosis. Then, based on the method of model analysis, the fault-tolerant control of the ECAS system under fault is successfully carried out by closing the air spring solenoid valve or adjusting it online.

The designed observer, controller, and vehicle model are run on the simulation platform. The validity of the above methods is verified by comparing the relationship between detection modes and adaptive thresholds under the four proposed fault behaviors. Finally, in order to verify the control effect on the actual vehicle, an HiL semi-physical test platform was built. Such a test platform can combine the actual controller with the simulation model. The test results prove that the fault diagnosis and fault-tolerant control methods proposed in this study can be used in actual controllers. At the same time, it can accurately diagnose the location and type of faults, so as to carry out effective active fault-tolerant control.

**Author Contributions:** Conceptualization, X.J. and X.X.; methodology, H.S.; software, X.J. and H.S.; validation, H.S., X.J. and X.X.; investigation, X.J. and X.X.; data curation, H.S.; writing—original draft preparation, X.J.; writing—review and editing, H.S. and X.J.; supervision, X.X.; project administration, X.X.; funding acquisition, X.X.; All authors have read and agreed to the published version of the manuscript.

**Funding:** This work is supported by the National Natural Science Foundation of China (Grant No. 51875256).

**Data Availability Statement:** Not applicable.

**Conflicts of Interest:** The authors declare no conflict of interest.

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
