# Peer review of "Model-Based Fault Diagnosis of Actuators in Electronically Controlled Air Suspension System"

_wevj, doi:10.3390/wevj13110219_

Round 1

Reviewer 1 Report

The article proposes a Model-based Fault Diagnosis of Actuators in ECAS System. The presented topic is up-to-date, especially for the economic reasons and the practical applications. The early damage of the system allows to guarantee the greater safety and the driving comfort. The research methodology described in the paper contains the required information as well as the research area. The study clearly outlines the purpose and the scope of the research. The work is well organized with an introduction, the theoretical part, the presentation of the results with conclusions. After the theoretical part, which presents the purpose of the research and their methodology, the analysis of the obtained results is presented. Each part is presented correctly and is correlated with the rest of the article.

Recommendations for improving the manuscript:

1. The description of the tables, especially the table 3 is poor - please complete it in the text of the article.

2. On what basis the times are defined in the text (eg The fault detection time is 11.1s, 10.4s and 10.1s, respectively - page 12).

3. Please complete the details about the site (chapter 5.1 - eg what is NI VeriStand software).

4. How many studies have been done, are the presented results supported by only one study?

5. Was the method tested on a real object?

6. Can the presented results be used in practice?

Author Response

Reply to the comments of Reviewer 1

Comment 1: ( “The description of the tables, especially the table 3 is poor - please complete it in the text of the article.” )

Reply to comment 1:

Thank you very much for pointing out the problems here. After reexamining the table annotations and related paragraph descriptions, we found the table was really difficult to understand. We have partially revised the table and added descriptions. The specific changes are as follows:

Table 3. Description of fault behavior.

Fault Number

1

2

3

4

Moment of failure /s

8

Fault behavior

0

0.2

0.4

0.6

0

0

0

0

To verify the proposed fault diagnosis program, four fault behaviors shown in Table 3 are selected. Fault 1 is a stuck fault, and faults 2-4 are constant gain faults. In the fault behavior,  represents the gain coefficient, and  represents the fault stuck value. The flow area of the valve port can be calculated from Equation (8). Therefore, the fault behavior represents the flow area of the valve port, corresponding to four fault behaviors. The actuators 1-4 respectively represent the front left, rear left, front right, and rear right air spring solenoid valves. The fault diagnosis module was started at the same time as the vehicle height adjustment was started in the 5s, and the fault occurred in the 8s. The first 5s is the process of the air spring model in AMEsim gradually returning to a steady state. The simulation results of faults1 and 2 detections are shown in Figure 6 and 7.

Comment 2: ( “On what basis the times are defined in the text (eg The fault detection time is 11.1s, 10.4s and 10.1s, respectively - page 12)” )\

Reply to comment 2:

It should be explained to you that the fault detection time indicated here is not the time set manually. In fact, they are the time when ECAS system successfully detects the fault and judges the fault location and type after adopting fault diagnosis. They depend on the time when the fault occurs and the response speed of the fault diagnosis system.

Comment 3: ( “Please complete the details about the site (chapter 5.1 - eg what is NI VeriStand software).” )

Reply to comment 3:

I am very willing to take your advice. The introduction of NI has been added in the article. In addition, we have added a diagram to introduce the operation principle of HiL system in detail. The modifications are as follows:

NI VeriStand software is a software environment for configuring real-time test applications. Its functions include configuring the operating system, board, vehicle model, input/output interface with the actual controller, and user interface. NI VeriStand software carries out hardware in the loop simulation by interacting with the signal of the actual controller. Researchers can monitor the simulation process in real time through the upper computer. The test principle of HiL platform is shown in Figure 12.

Figure 12. The working principle of HiL test platform

Comment 4 ( “How many studies have been done, are the presented results supported by only one study?” )

Reply to comment 4:

This research is not a single achievement. In fact, this is a part of a research project on the dynamic characteristics mechanism and control of vehicle electric control air suspension system. The research on fault diagnosis and fault-tolerant control of ECAS system not only focuses on actuator faults in this paper, but also considers sensors. In addition, we also try to use other active fault-tolerant control methods to improve the effect. More research results may be released in the future.

Comment 5 ( “Was the method tested on a real object?” )

Reply to comment 5:

The fault diagnosis and fault-tolerant control methods proposed in this study have all been run on the physical controller. This allows us to study whether the proposed method can meet the response time and control effect of the actual controller. We run the proposed method on the virtual simulation environment and HiL hardware in the loop test platform respectively. These two verification environments have confirmed the effectiveness of the method.

Comment 6 ( “Can the presented results be used in practice?” )

Reply to comment 6:

Thank you for your questions. This research has verified the effectiveness of the proposed method on the physical controller. Other parts, including the vehicle model, are set up in the NI real-time simulator. This HiL hardware in the loop simulation platform allows us to focus on the key points we need to study (fault diagnosis and fault-tolerant control methods). In the future research, we will consider applying the research results to real vehicles to obtain more research results. Thank you again for your suggestions.

Reviewer 2 Report

The contribution of this paper is to use the extended Kalman filter bank based on adaptive threshold to study the fault diagnosis and fault-tolerant control of air spring solenoid valve.

Mark or highlight important elements in Figure 1.

The quality of the pictures and description is good.

This article has conducted research on the fault diagnosis and fault-tolerant control of ECAS system actuators.

I have to commend the authors that diagnostics are proposed, simulation processes are shown and performed and Finally, a HiL simulation monitoring platform was built.

In my opinion, the whole system could work, so it is realistic. The article is written according to the template and fulfills all the conditions to be published in the given magazine. I want to highlight the quality of the article and also the form. I recommend publishing an article in a given magazine.

Author Response

Reply to the comments of Reviewer 2

Comment 1 ( “Mark or highlight important elements in Figure 1.” )

Reply to comment 1:

Thank you very much for your affirmation of our research work. Your suggestion is very meaningful, and we are very willing to accept this change. In this study, the most influential component of ECAS system failure is the solenoid valve of air spring. We readjusted the shape and color of the mark where the air spring solenoid valve is located to highlight its importance. This change can help readers understand the research object of the article more quickly. The changes are as follows:

Comment 2 ( “The quality of the pictures and description is good.” )

Reply to comment 2:

Thank you very much for your affirmation of the picture description and quality of this article. Clear pictures and accurate descriptions can make articles more readable. Therefore, we attach great importance to the presentation effect in this aspect.

Comment 3 ( “This article has conducted research on the fault diagnosis and fault-tolerant control of ECAS system actuators.” )

Reply to comment 3:

Thank you for your careful reading. Your evaluation here is very accurate. This paper focuses on the fault diagnosis and fault-tolerant control of ECAS system under actuator fault. An accurate mathematical model is established according to the fault types. The extended Kalman filter bank with adaptive threshold is used for fault diagnosis. After using the method of model analysis to locate and judge the fault, fault tolerant control is completed by closing or online adjusting the air spring solenoid valve.

Reviewer 3 Report

The authors presented Model-based Fault Diagnosis of Actuators in ECAS System. Overall, the article is well-written. I have a few comments.

1. Literature review is short. Add the recent articles and then discuss about the research gaps.  

2. Increase the font size of all figures. At present, they are hardly readable. 

3. The authors need to justify the proposed method by comparing with the recent works. 

4. Conclusion is too descriptive. Try to summarize your key findings. 

5. Try to avoid in the abbreviation in the title. Use the full name of " ECAS". 

Author Response

Reply to the comments of Reviewer 3

Comment 1 ( “Literature review is short. Add the recent articles and then discuss about the research gaps.” )

Reply to comment 1:

Your suggestions are very informative. We decided to analysis the gap with the existing research after the literature review to highlight the necessity of this study. And contributions and shortcomings of present technology will be described in introduction. Finally, we will replace and add more timely references. The revised contents are as follows.

Air suspension can improve vehicle ride comfort and road friendliness, and its natural frequency is low and has variable stiffness characteristics [1 -5]. But the general air suspension cannot adjust the suspension stiffness and damping according to the load change. The natural frequency and controllability of the electronically controlled air suspension (ECAS) is low, which can further improve the vehicle ride comfort and control stability [6-8]. In recent years, researches on ECAS mainly focus on improving comfort and stability. In 2019, Rui modeled the ECAS system according to its nonlinear characteristics and designed an adaptive sliding mode control strategy. The method effectively improves the stability of the system under considering the parameter uncertainty [9]. In 2021, Ma et al. designed an integrated control strategy to solve the problems of small stiffness adjustment range and poor roll stability of traditional ECAS systems. The handling stability and anti roll performance of the vehicle are obviously improved [10]. In 2021, Hu et al. conducted research on the hybrid control of body height and attitude of ECAS system. They built a vehicle model based on mixed logic dynamics and designed the switching strategy of solenoid valve. The coordinated control between ECAS system body height and attitude is well solved, and good vibration isolation performance and stability are achieved [11].

  • Rui B. Nonlinear adaptive sliding-mode control of the electronically controlled air suspension system [J]. International Journal of Advanced Robotic Systems, 2019, 16(5): 443-475.
  • Ma, Y.; Yan, T.; Zhao, Y. Research on Integrated Control Strategy of a New-Type Electronically Controlled Air Suspension System[J]. Automotive Engineering, 2021, 43(9): 1394-1401.
  • Hu, Q.; Lu, W.; Jiang, J. Design of a vehicle height and body posture adjustment hybrid automaton of electronically controlled air suspension[J]. International Journal of Adaptive Control and Signal Processing, 2021, 35(9): 1879-1897.

At present, the research on fault diagnosis and fault-tolerant control of ECAS system is less. In addition, the research on ECAS vehicle height control rarely focuses on how to ensure effective control in case of actuator failure. However, the ECAS system with actuator failure has many problems to be solved, such as risk analysis, fault diagnosis architecture and fault tolerant control strategy design.

Comment 2 ( “Increase the font size of all figures. At present, they are hardly readable.” )

Reply to comment 2:

We are happy to accept this proposal. Now most of the pictures have been modified and the character size has been adjusted. I hope all readers can see the information contained in the pictures more clearly.

Comment 3 ( “The authors need to justify the proposed method by comparing with the recent works.” )

Reply to comment 3:

Your suggestions are very useful. We have listed the methods and contributions of recent research, and described the principles and advantages of the methods used in this study. At the same time, some references are updated. The revised contents are as follows.

The fault location and type can be determined by fault diagnosis. In this way, fault tolerant control (FTC) can be implemented. Alwi et al. classified fault-tolerant control methods in detail [20]. Fault tolerant control is usually divided into passive FTC (PFTC) and active FTC (AFTC). The common methods of passive fault-tolerant control include H methods based on \infty control theory [21] and sliding mode control theory [22]. The characteristic of passive fault-tolerant control is that there is no need for fault diagnosis, and the controller parameters are not changed, so it is easy to implement, but the fault-tolerant control is limited. The active fault-tolerant control adjusts the controller parameters on line or configures the controller structure units on line based on the fault diagnosis information to realize the system stability which is different from the passive fault-tolerant control. Active fault-tolerant control methods can be divided into two types: planning type and on-line adjustment type [23]. In the planning FTC, the controller is designed in advance for all possible faults of the system, and the corresponding controller is activated when the corresponding fault occurs. On the other hand, the online adjustment of the controller is mainly through adaptive control or control signal redistribution to achieve fault-tolerant control [24,25]. In 2020, Pang et al. designed a fault-tolerant controller based on the nonlinear suspension system, which combines the state feedback observer and the H-infinity observer. The purpose of fault-tolerant control is achieved by comparing the system state under the fault free state to compensate [26] In 2021, Xue et al. considered the failure of 1/4 active suspension actuator under the change of sprung mass. They use the finite element neural network method to approximate the observer fault, and turn it into a linear matrix inequality problem. This method can get the results faster and more accurately, and can well adapt to the changing spring mass [27]. The above methods are very effective, but the actuator fault diagnosis and fault-tolerant control for ECAS system are rarely used. In order to fill the shortage in this field, this research will use the extended Kalman filter bank with adaptive threshold as the observer, and use the online adjustment method to continuously improve the body height. Compared with the existing research, this method has stronger adaptability and accuracy.

  • Pang, H.; Liu, X.; Shang, Y.; Yao, R. A hybrid fault-tolerant control for nonlinear active suspension systems subjected to actuator faults and road disturbances [J]. Complexity, 2020, 2020(23): 1-14.
  • Xue, W,; Jin, P.; Li, K. Parameter-dependent actuator fault estimation for vehicle active suspension systems based on RBFNN:[J]. Proceedings of the Institution of Mechanical Engineers, Part D: Journal of Automobile Engineering, 2021, 235(9):2540-2550.

Comment 4 ( “Conclusion is too descriptive. Try to summarize your key findings.” )

Reply to comment 4:

We share your views on this issue. The conclusion part describes the work content of this study in too much detail, but does not clearly describe the conclusions of each work. Therefore, we have simplified the description of the conclusion and highlighted the key findings. The revised contents are as follows.

In this paper, the fault diagnosis and active fault-tolerant control of ECAS system under actuator fault are studied. Based on the fault model, an extended Kalman filter bank with adaptive threshold is designed for fault diagnosis. In addition, online adjustment is adopted for fault-tolerant control.

Firstly, the ECAS vehicle model is simplified, and the mathematical model of vehicle height regulation is established as the basis of fault tolerant control. The faults are classified into constant gain faults and stuck faults by fault tree method. According to the fault type, an accurate fault mathematical model is established. Then, an adaptive threshold extended Kalman filter bank is designed as the observer. Each residual is compared with the adaptive threshold. Therefore, the fault location and type can be accurately located. This method improves the accuracy and speed of diagnosis. Then, based on the method of model analysis, the fault-tolerant control of the ECAS system under fault is successfully carried out by closing the air spring solenoid valve or adjusting it online.

The designed observer, controller and vehicle model are run on the simulation platform. The validity of the above methods is verified by comparing the relationship between detection modes and adaptive thresholds under the four proposed fault behaviors. Finally, in order to verify the control effect on the actual vehicle, a HiL semi physical test platform was built. Such a test platform can combine the actual controller with the simulation model. The test results prove that the fault diagnosis and fault-tolerant control methods proposed in this study can be used in actual controllers. At the same time, it can accurately diagnose the location and type of faults, so as to carry out effective active fault-tolerant control.

Comment 5 ( “Try to avoid in the abbreviation in the title. Use the full name of " ECAS".” )

Reply to comment 5:

Thank you for your suggestion. After consideration, we decided to change the title of the lecture to ‘Model-based Fault Diagnosis of Actuators in Electronically Controlled Air Suspension System’.
